# Asymptomatic Left Ventricular Hypertrophy Is a Potent Risk Factor for the Development of HFpEF but Not HFrEF: Results of a Retrospective Cohort Study

**DOI:** 10.3390/jcm11133885

**Published:** 2022-07-04

**Authors:** Artem Ovchinnikov, Evgeny Belyavskiy, Alexandra Potekhina, Fail Ageev

**Affiliations:** 1Out-Patient Department, Institute of Clinical Cardiology, National Medical Research Center of Cardiology Named after Academician E.I. Chazov, 3-d Cherepkovskaya St., 15a, 121552 Moscow, Russia; potehina@gmail.com (A.P.); ftageev@gmail.com (F.A.); 2Department of Clinical Functional Diagnostics, A.I. Yevdokimov Moscow State University of Medicine and Dentistry, Delegatskaya St., 20, p. 1, 127473 Moscow, Russia; 3Department of Internal Medicine and Cardiology, Charité—Universitätsmedizin Berlin, Campus Virchow—Klinikum, Augustenburger Platz, 1, 13353 Berlin, Germany; evgeny.belyavskiy@charite.de

**Keywords:** hypertension, left ventricular hypertrophy, echocardiography, diastolic dysfunction, heart failure with preserved ejection fraction, heart failure with reduced ejection fraction

## Abstract

(1) Background: The structural and functional features of the natural history of asymptomatic hypertensive left ventricular hypertrophy (LVH) are not clearly defined. (2) Objective: To determine structural and functional changes in asymptomatic hypertensive LVH, as well as the incidence and predictors of the transition to different phenotypes of heart failure (HF) after a long-term follow-up. (3) Methods: Based on the assessment of chart reviews, we retrospectively selected 350 asymptomatic patients with hypertensive concentric LVH and LV ejection fraction (EF) ≥ 50%. After a median follow-up of 8.1 years, 223 patients had a re-assessment. The final diagnosis (HF with reduced EF [HFrEF], or HF with preserved EF [HFpEF]) was established according to current recommendations. (4) Results: After a follow-up, only 13% of patients remained asymptomatic, 72% developed HFpEF, and 15% developed HFrEF. The transition to HFpEF was associated with an increase in LV diastolic dysfunction grade in 62% of patients. Multivariable analysis identified age, duration of hypertension, interval changes in LV mass, and a lack of statin treatment as independent predictors of HFpEF. Among 34 patients who developed HFrEF, 16 patients (7% of the whole group) had no interval myocardial infarction, corresponding to an internal mechanism of systolic dysfunction. All these 16 patients had mild systolic dysfunction (LVEF > 40%). Baseline LVEF and LV end-diastolic dimension, and interval atrial fibrillation were identified as predictors of internal HFrEF. (5) Conclusions: The majority of patients with asymptomatic LVH developed HFpEF after long-term follow-up, which was associated with the deterioration of LV diastolic dysfunction and a lack of statin treatment. In contrast, the transition to HFrEF was infrequent and characterized by mild LV systolic dysfunction.

## 1. Introduction

Left ventricular hypertrophy (LVH) is a common structural cardiac alteration that may be a physiological adaptation to exercise or a pathological condition that is either genetic or secondary to LV overload. Physiological LVH is usually benign and regresses upon the cessation of physical activity. Pathological LVH is maladaptive and evolves towards progressive LV dysfunction.

The most common cardiovascular condition associated with LVH is arterial hypertension [1]. In a pooled analysis of studies utilizing echocardiography for the detection of LVH, the reported prevalence of LVH ranged from 36% to 41% among patients with hypertension [2]. Arterial hypertension is characterized by the development of a concentric pattern of LVH as a result of the increased LV systolic pressure and afterload. Hypertensive LVH is a potent cardiovascular risk factor independent of the degree of blood pressure (BP) elevation or other comorbidities and correlates with biomarkers [3,4]. However, little is known about the structural and functional features of the natural course of hypertensive LVH, or hypertensive heart disease (HHD). HHD encompasses a broad clinical spectrum ranging from clinically silent structural concentric remodeling to the development of clinical symptoms—often decades later—such as heart failure (HF). Arterial hypertension is the most prevalent risk factor for the development of HF [5]. The onset of symptomatic HF is an important indicator of poor outcomes and a high mortality rate, both in HF with preserved ejection fraction (HFpEF) and HF with reduced ejection fraction (HFrEF) [6].

According to the classic paradigm, concentric LVH is a compensatory mechanism aimed at adapting to higher demands for LV work, including pressure load [7]. Subsequently, after a series of poorly characterized events (“transition to failure”), the left ventricle may dilate with EF drop (HFrEF-phenotype), or develop concentric remodeling, progressive LV diastolic dysfunction (LVDD), and increased filling pressures but preserved EF (HFpEF-phenotype) [8]. The threshold between adaptive (healthy) and maladaptive (pathologic) hypertrophy, as well as the determinants of clinical and structural deterioration in LVH, are not clearly defined, highlighting some unresolved controversies. We aimed to determine structural and functional changes in asymptomatic hypertensive concentric LVH, as well as the incidence and predictors of the transition to different HF phenotypes (HFpEF vs. HFrEF) during a retrospective cohort study.

## 2. Materials and Methods

### 2.1. Study Population

The outpatient chart reviews of patients that visited the Out-Patient Department of the National Medical Research Center of Cardiology in Moscow (Russian Federation) between January 2002 and September 2010 and their corresponding reports from transthoracic echocardiograms were screened. The baseline collection of data on the index study (clinical status, concomitant diseases, echocardiography data, etc.) was carried out according to archival documents, where they were recorded regardless of the aims of the present study.

Asymptomatic hypertensive patients aged ≥ 18 years with concentric LVH (LV mass index > 115 g/m^2^ in men and >95 g/m^2^ in women and relative wall thickness [RWT] > 0.42) and normal LVEF (≥50%), i.e., corresponding HHD class IIA [9], were identified. An asymptomatic course of LVH was evidenced by the absence of any exercise limitation and no more than LVDD grade I as noted in chart reviews and echocardiography reports, respectively. The inclusion of patients with chronic atrial fibrillation (AF) was allowed provided that AF was completely asymptomatic. Those patients with alternative causes of LVH, asymmetrical wall hypertrophy, eccentric LVH, symptomatic HF, advanced LVDD (grades II-III), secondary hypertension, significant valve disease, known unrevascularized coronary artery disease, and LV dilatation (LV end-diastolic dimension [EDD] ≥ 5.9 cm in men and ≥5.3 cm in women) were excluded. The study protocol was approved by the Independent Ethic Committee for Clinical Trials of the National Medical Research Center of Cardiology (permission no. 190).

### 2.2. Study Design

Based on the assessment of chart reviews, we retrospectively identified a total of 350 subjects meeting inclusion/exclusion criteria. Initial patient demographic and clinical data were revealed from an outpatient chart review, including the electrocardiographic, radiographic, and echocardiographic reports generated for clinical purposes. After a median follow-up of 8.1 (6.9–10.1) years (with a minimal follow-up of at least 5 years), 27 patients (12%) died. The death occurred due to cardiovascular causes in 17 patients (MI in seven, congestive HF in four, sudden cardiac death and stroke each occurring in three), and noncardiovascular causes in four patients; in six patients the cause of death was unknown. All patients who died of cardiovascular causes had moderate/severe LVH (LVMi ≥ 132 g/m^2^ in men and ≥109 g/m^2^ in women) at baseline; nine (53%) were men.

Surviving patients were invited to the follow-up visit; 100 of them refused to participate in the study or were lost to follow-up; thus, the follow-up study was performed on 223 patients (64%) (Figure 1). Given that the exact cause of death was unknown in 6 of 27 deceased patients (22%), and because we did not know the exact number of patients who died, as some patients were lost to follow-up, the effect of LVH on the cardiovascular mortality was behind the scope of the present study. Informed consent was obtained from all individual participants included in the study which was conducted in accordance with the Declaration of Helsinki.

The follow-up assessment included meticulous clinical evaluation, echocardiography, and N-terminal pro-B-type natriuretic peptide (NT-proBNP) blood level analysis.

During the follow-up study, HFpEF was diagnosed according to the current recommendations as follows: symptoms and/or signs of HF, preserved LVEF (≥50%), relevant structural heart disease (LVH or increased left atrial [LA] volume index > 34 mL/m^2^) or functional (average E/e′ ≥ 13 and/or average septal and lateral e′ velocity < 9 cm/s) abnormalities with an additional increase in NT-proBNP level > 125 pg/mL [10]. If the levels of natriuretic peptides were not increased, the elevated LV filling pressures were evidenced at rest (LVDD grades II–III) or during the diastolic stress-test. HF with reduced EF was diagnosed when the patient had symptoms and/or signs of HF, and LVEF < 50% [10].

When a decline of LV ejection fraction was observed, the mechanism of contractile deterioration was identified as external if the clinical history or instrumental data indicated interval myocardial infarction [MI], or as internal (due to the long-standing LV pressure overload) in all other cases.

### 2.3. Echocardiography

An echocardiographic assessment was performed using ultrasound systems HDI 5000 (Philips, Bothell, WA, USA) and iE33 (Philips, Andover, MA, USA) ultrasound machines by experienced cardiac sonographers.

Wall thickness, chamber volumes, and LVEF were determined in accordance with the current guidelines [11]. The measurement of LV mass was performed using the M-mode method by Devereux et al. [12] and indexed to body surface area. LVH was defined as LV mass index (LVMi) >115 g/m^2^ in men and >95 g/m^2^ in women.

The relative wall thickness (RWT) was defined as (septal wall thickness + posterior wall thickness)/LVEDD with the further categorization of an increase in LVMi as either concentric (RWT > 0.42) or eccentric (RWT ≤ 0.42) hypertrophy [11]. LV systolic function was considered preserved at ≥50% or reduced (mildly, moderately, or severely)at 40–50%, 30–39%, and <30%, respectively.

Both at baseline and at follow-up, LVDD grade was assessed by measuring the mitral inflow velocities (E, A), LA diameter or volume, pulmonary venous flow velocities (peak systolic and diastolic velocities and their ratio [S/D]), and pulmonary artery systolic pressure (PASP); some diastolic parameters, such as mitral annulus relaxation velocity (mitral e′) and mitral E/e′ ratio, were routinely measured only at the follow-up examination. Elevated LV filling pressure at rest was verified if LVDD of grade II–III was revealed, and at exercise (during supine bicycle exercise)—if exercise-induced elevation in E/e′ (average E/e′ > 14) and tricuspid regurgitation velocity > 2.8 m/s were observed [13]. PASP was calculated as a sum of peak tricuspid regurgitation and right atrial pressure estimated by inferior vena cava size and its collapse.

All measurements represent the mean of ≥3 beats.

### 2.4. NT-proBNP

Plasma level of the myocardial stress marker N-terminal pro–brain natriuretic peptide (NTproBNP) was measured only during the follow-up study via automated electrochemiluminescence immunoassay (Roche Diagnostics, Mannheim, Germany). The detection limit of the NTproBNP assay was 5 pg/mL.

### 2.5. Study Endpoint

The primary endpoint was the development of a clinical HF (with reduced vs. preserved LVEF) on a follow-up assessment.

### 2.6. Statistical Analysis

Statistical analysis was performed using standard software (MedCalc, version 19.5.3, Ostend, Belgium). Data are presented as the median (interquartile range); categorical variables are reported as numbers and percentages of observations. The Wilcoxon test was applied to the change from baseline. The differences in parameters at baseline and after the follow-up between two different groups were tested using the Mann–Whitney U test, and the χ^2^ test, or the Fisher’s exact test for qualitative data.

The relationship between binary variables and categorical or continuously distributed variables was analyzed with logistic stepwise regression. A value of *p* < 0.05 was considered statistically significant.

## 3. Results

### 3.1. Patient Baseline Characteristics

Data presented here were derived retrospectively from chart reviews. The mean age of participants was 59 years, and 65% were men; all were Caucasians. The study subjects had long-term arterial hypertension (the median duration was 20 [11–30 years]) complicated by asymptomatic concentric LVH. They were mainly obese with multiple comorbidities, including ischemic heart disease, diabetes, and chronic kidney disease (Table 1). The majority of patients (90%) had mild LVDD (grade I), which is associated with normal LV filling pressures at rest; another 10% had asymptomatic chronic AF.

### 3.2. Patient Follow-Up Characteristics

After a median follow-up of 8.1 years, only 28 (13%) patients remained asymptomatic; 161 patients (72%) developed HFpEF, and 34 patients (15%) developed HFrEF. Interval transmural MI was associated with the development of HFrEF in 18 patients (8% of the whole group), and prolonged LV pressure overload in 16 patients (7% of the whole group, Figure 2).

After the follow-up, the proportion of patients with chronic AF, MI, diabetes, and chronic kidney disease increased (Table 1).

In the follow-up assessment, a worsening in LVDD (grade increase from I to II-III) was revealed in more than half of the patients (52%), and the proportion of patients with chronic AF increased from 10% to 18% (*p* = 0.027, Figure 2). The progression of LVDD was accompanied by a highly significant increase in the LA size and PASP compared with baseline values (*p* < 0.001 for both variables). Pulmonary hypertension (PASP > 35 mm Hg) was detected in 16% of patients at baseline and in 44% of patients at follow-up (*p* < 0.001, Table 1).

LVMi did not change in the total study population (*p* = 0.17), but a significant increase in RWT was observed (*p* < 0.001, Table 1). Only five patients (2%) moved from concentric to eccentric LVH; four of them had external and one had internal LV systolic dysfunction. A patient was considered to have received antihypertensive treatment if the duration of the therapy exceeded half of the time elapsed between the studies and at least one year prior to the follow-up study. During the follow-up, two-thirds of the patients received ACE inhibitors/angiotensin receptor blockers, and more than half of the patients received beta-blockers; 38%—diuretics, and a quarter of the patients—calcium channel blockers. Most of the patients (62%) received combined antihypertensive therapy consisting of two (1–3) drugs. Forty percent of participants received statins.

### 3.3. The Comparison of Patients with New-Onset Heart Failure and Those Who Remained Asymptomatic after the Follow-Up

The baseline and follow-up clinical characteristics, stratified by the progression to HF are listed in Table 2.

Patients who developed HFpEF were older compared with patients who remained asymptomatic. Patients with internal HFrEF were also older than asymptomatic patients (the median age at baseline was 54 and 58 years, respectively); however, due to the small number of both subgroups, the difference was not significant.

Both HF subgroups had a longer history of arterial hypertension, less frequently received statins, and more frequently received loop diuretics than patients who remained asymptomatic. All three subgroups were comparable in baseline blood pressure and its interval changes, as well as in the interval antihypertensive therapy and the average level of low-density lipoprotein (LDL) cholesterol (Table 2).

At baseline and follow-up studies, there were no differences in the incidence of comorbidities between all three subgroups. Chronic AF was more common in patients with internal HFrEF both at baseline and at follow-up than in patients who developed HFpEF (Table 2). None of the asymptomatic patients had chronic AF. All three subgroups did not differ in body mass index (BMI) at baseline; at least half of the patients in each subgroup were initially obese. However, only patients with internal HFrEF showed a significant increase in BMI at follow-up (*p* = 0.040 compared with baseline). At baseline, patients in sinus rhythm from all three subgroups had mild LVDD (grade I). At follow-up, there were no asymptomatic patients who increased LVDD grade, while 62% of patients with HFpEF and 67% of patients with internal HFrEF increased LVDD (*p* < 0.01 for both comparisons vs. asymptomatic patients). This difference was accompanied by a more significant interval increase in LA size and PASP (Table 3). The HF subgroups were comparable in follow-up E/e′ ratio (a marker of LV filling pressure) and significantly outperformed asymptomatic patients (*p* < 0.01 for both comparisons; Table 3). In the follow-up study, a larger proportion of patients with internal HFrEF had pulmonary hypertension (69%) compared with HFpEF patients (42%) and asymptomatic patients (7%, *p* < 0.05 for both comparisons). Patients with HFpEF had higher follow-up NT-proBNP levels (262 (171–489) pg/mL) than asymptomatic patients (102 (73–137) pg/mL, *p* < 0.001), but less than patients with internal HFrEF (551 (311–1400) pg/mL, *p* < 0.01).

At baseline, all three subgroups did not differ in the severity of LVH; however, asymptomatic patients had a lower median LVMi (118 g/m^2^) than patients who developed HFpEF (136 g/m^2^, *p* = 0.059) or internal HFrEF (156 g/m^2^, *p* = 0.067).

At follow-up, LVMi significantly decreased in asymptomatic patients (by 14%, *p* = 0.03), and non-significantly increased in patients with HFpEF (by 5%, *p* = 0.068) and in patients with internal HFrEF (by 4%, *p* = 0.23), resulting in a significant difference in LVMi dynamics between the asymptomatic subgroup and both HF subgroups (*p* < 0.05 for both comparisons).

Patients with internal HFrEF had a higher baseline LVEDD, and a lower LVEF and RWT compared with asymptomatic patients or patients with HFpEF (*p* < 0.05 for all comparisons; Table 3). After a follow-up, RWT significantly increased in all three subgroups, with a greater interval increment in HFpEF patients compared with asymptomatic patients (*p* = 0.015). All asymptomatic and HFpEF patients retained concentric LVH; only one of 16 patients with internal HFrEF had a transition to eccentric LVH. None of the patients with internal HFrEF developed severe LV systolic dysfunction (i.e., LVEF < 30%).

### 3.4. The Predictors of HF Development

Using multivariate stepwise logistic regression analysis, the independent predictors of the development of HFpEF were identified among patients with concentric LVH. Since the transition from asymptomatic LVH to HFpEF is obviously accompanied by worsening LVDD, the variables reflecting the diastolic deterioration, such as an increase in the LVDD grade, interval changes in the LA size, or PASP were not included in the analysis.

The age, duration of hypertension, interval changes in LVMi, and statin treatment were significantly associated with the development of HFpEF (Table 4), with statin treatment having the strongest odds ratio (0.31 [95% CI: 0.13 to 0.76]).

Similarly, predictors of the development of internal HFrEF were determined. Baseline LVEF, baseline LVEDD, and interval AF were significantly associated with the development of internal HFrEF (Table 5), with interval AF having the strongest odds ratio (6.4 [95% CI: 1.33 to 17.6]).

## 4. Discussion

LVH is an important event in the progression of HHD and is associated with an increased risk of adverse outcomes including HF development [6]. In the present retrospective cohort study, we assessed the structural and functional features of the natural history of asymptomatic hypertensive LVH after a median follow-up of 8 years. The majority of asymptomatic LVH patients developed HFpEF as a result of LVDD deterioration, while the transition to HFrEF was rare (less than 1% per year).

The participants predominantly demonstrated LVDD worsening: an increase in diastolic dysfunction grade occurred in 62% of patients who developed HFpEF, and none of the asymptomatic patients. The progression in LVDD in patients with HFpEF was associated with a larger increment in RWT compared with asymptomatic subjects. An increase in thickness/dimension ratio reduces LV chamber distensibility and may impair LV twist-untwisting [14] and diastolic suction—an important mechanism facilitating early diastolic filling.

According to the novel HFpEF paradigm, chronic myocardial inflammation is the main pathophysiologic mechanism of diastolic deterioration in patients with HHD. Proinflammatory comorbidities, such as hypertension, metabolic disorders, diabetes mellitus, and renal insufficiency trigger a low-grade systemic inflammatory state and coronary microvascular endothelial dysfunction with subsequent cardiomyocyte hypertrophy, myocardial infiltration with activated leukocytes and cardiac fibrosis [15]. Although the inflammation status was not assessed in the present study, we suggest that chronic inflammation played an important role in the transition from asymptomatic LVH to HFpEF in the study participants, since the use of statins prevented this transition. The anti-inflammatory and antifibrotic effects of statins are well-documented [16], these effects may mediate the beneficial action of statins on diastolic function. Statins improved LV diastolic function by affecting inflammatory and fibrotic cytokine networks in experimental studies [17,18,19]. A positive effect of statins on diastolic function has been shown in several cardiovascular populations, such as in patients with coronary artery disease [20], or hyperlipidemia [21]. A meta-analysis of 11 observational studies showed the use of statins was associated with a 40% reduction in mortality in patients with HFpEF [22].

To date, a wide range of pharmacotherapies has demonstrated minimal impact on outcomes in HFpEF. As morbidity and mortality associated with HFpEF continue to escalate, the focus on its prevention is increasingly important. The systemic inflammation as a “trigger” for the development of HFpEF suggests the use of statins not only in patients with clinically obvious HFpEF but also in asymptomatic patients with a high risk of HF, including patients with compensated LVH. In the present study, patients who remained asymptomatic after the follow-up showed a significant increase in LA size and RWT, which clearly indicates a very high risk of HF. Early statin therapy could suppress myocardial microvascular inflammation and diminish the progression of LVDD. According to our results, we speculate that statins predominantly demonstrate preventive rather than treatment effects in patients with compensated LVH.

Lipid-lowering therapy continues to be a challenge in the treatment of cardiovascular diseases, including LVH and HFpEF, as statins are the group of drugs that are most commonly underused and underdosed [23]. In the present study, only 34% of patients with HFpEF were receiving statins, although most of these patients were at high or very high cardiovascular risk and candidates for lipid-lowering therapy.

Asymptomatic patients and patients who developed HFpEF were comparable in the baseline LVH but differed in LVH dynamics during the follow-up, and the increase in LVMi was a significant predictor for the transition from asymptomatic LVH to HFpEF. Despite the differences in LVH dynamics, asymptomatic and HFpEF patients were comparable in the severity of hypertension and antihypertensive therapy. However, asymptomatic patients were younger and had a shorter history of hypertension. These differences could indicate greater plasticity of hypertrophy and its predisposition to regress.

The relationship between hypertension and LV mass is complex as patients with hypertension and concentric remodeling usually have comorbidities and advanced age that independently affect LVH and/or LVDD [24,25,26]. In this study, asymptomatic patients had a lower incidence of comorbidities associated with concentric remodeling (coronary artery disease, diabetes and chronic kidney disease) than HFpEF patients, which could lead to different LVH dynamics despite a similar antihypertensive treatment.

All patients who developed HFpEF demonstrated concentric LVH after the follow-up. The larger clinical trials showed ≈50% of participants with HFpEF had LVH or concentric remodeling [27,28], although the prevalence of hypertension was ≈90% [29,30]. Many patients with a history of hypertension and without LVH have an abnormal diastolic function and apparent HFpEF, but a minority of patients with HFpEF show eccentric remodeling [31].

According to the classical paradigm of the natural course of LVH, which was presented 60 years ago [32] and confirmed by some experimental data [33,34], concentric LVH leads to impairments in LV contractility, which is associated with the depletion of myocardial adaptive reserves. Numerous studies have shown that despite overall preservation of EF, patients with LVH display subtle abnormalities in chamber and myocardial contractility [35,36,37]. Here, we demonstrated that LV systolic dysfunction due to long-lasting pressure overload (“intrinsic” mechanism) is a rare complication of a HHD, does not lead to a severe LV systolic dysfunction, and is not accompanied by a transition to eccentric LVH. These data are in line with the results of other studies [38,39,40] and generally confirm the compensatory nature of concentric LVH to overcome pressure overload and maintain LV pumping function. A decrease in LVEF was found in only 27% of patients who had a myocardial infarction. We speculate the concentric LVH may be protective against both “internal”, and post-infarction (“external”) systolic dysfunctions.

Study participants who developed internal HFrEF had the highest baseline LV size compared with the rest of the subjects, and LVEDD was a significant predictor of the development of internal HFrEF. The larger LV size may indicate a higher LV systolic wall stress and a greater vulnerability to pressure overload and predisposition to the development of systolic dysfunction.

Interval AF was another strong predictor of internal HFrEF, associated with a 6.4-fold increased risk of HFrEF. Important clinical sequela of AF is worsening HF due to loss of atrial systole in preload-dependent stiff left ventricle and deterioration of systolic function in persistently elevated heart rate (i.e., tachycardia-mediated cardiomyopathy) [41].

LV filling is mainly driven by high left atrial pressure during AF, but it’s not enough for adequate filling of the rigid hypertrophied ventricle. This leads to insufficient stretching of cardiomyocytes and a decrease in contractility (according to the Frank-Starling mechanism), but this deficit in LV filling is not sufficient to develop severe systolic dysfunction. In the present study, no patient with intrinsic HFrEF had a significant decrease in LVEF (<40%).

New-onset AF increases the risk of sudden cardiac death in patients with LVH [42]. Therefore, the prevention of AF could be a strategy to improve outcomes in HHD. This issue was addressed in two post hoc analyses of randomized trials: patients receiving angiotensin receptor blockers had a significant reduction in the relative risk of new-onset AF [43,44]. Another study demonstrated that HFpEF patients receiving stating were less prone to develop AF [45].

Twenty-seven patients died before finishing the follow-up; among those with an established cause of death, 81% were of cardiovascular disease, which highlights the fact that hypertensive LVH is a potent cardiovascular risk factor [3,4]. The Framingham Heart Study documented a significant relation between left ventricular mass and incidence of clinical events, including cardiovascular death [46], and all our patients who died of cardiovascular causes had advanced LVH.

### Study Limitations

The present study was retrospective and enrolled a relatively small group of patients. However, to our knowledge, the follow-up period was one of the longest among the studies with the comparable design. The severity of LVDD at baseline was assessed retrospectively when mitral e′ tissue Doppler was not used routinely, which could lead to some inaccuracy. However, mitral inflow, pulmonary venous flow velocities, LA size and PASP were determined in each patient, which helped to correctly assess the severity of LVDD [13].

The clinical assessment at baseline did not include the routine measurement of NT-proBNP levels. However, according to recent data, up to 20% of patients with invasively proven HFpEF have normal NT-proBNP; predominantly obese patients with concentric LVH [47], and elevated NT-proBNP is no longer a mandatory diagnostic criterion for HFpEF [47].

## 5. Conclusions

In the present retrospective cohort study, HFpEF developed in the majority of patients with asymptomatic LVH after a median follow-up of 8 years. The strongest independent risk factor for this transition was the lack of statin treatment, which might support the use of statins in patients with LVH. In contrast, LV systolic dysfunction due to prolonged pressure overload was a rare complication of hypertensive LVH. AF was a strong predictor of internal HFrEF, highlighting the role of preventive strategies to maintain sinus rhythm in patients with asymptomatic hypertensive LVH.

## Figures and Tables

**Figure 1 jcm-11-03885-f001:**
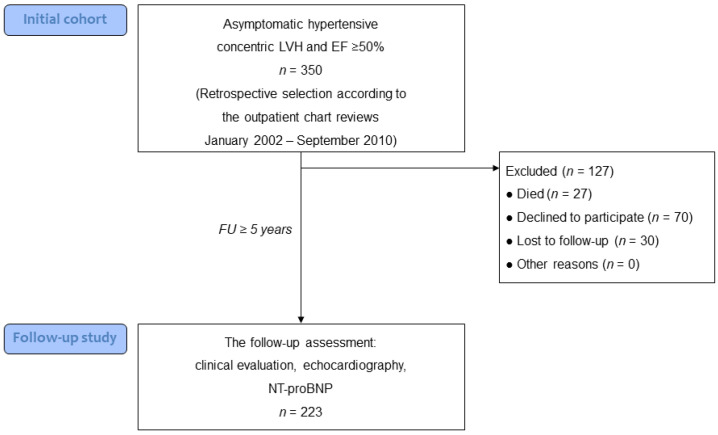
Flow chart of patient enrolment. EF indicates ejection fraction; FU, follow-up; LVH, left ventricular hypertrophy.

**Figure 2 jcm-11-03885-f002:**
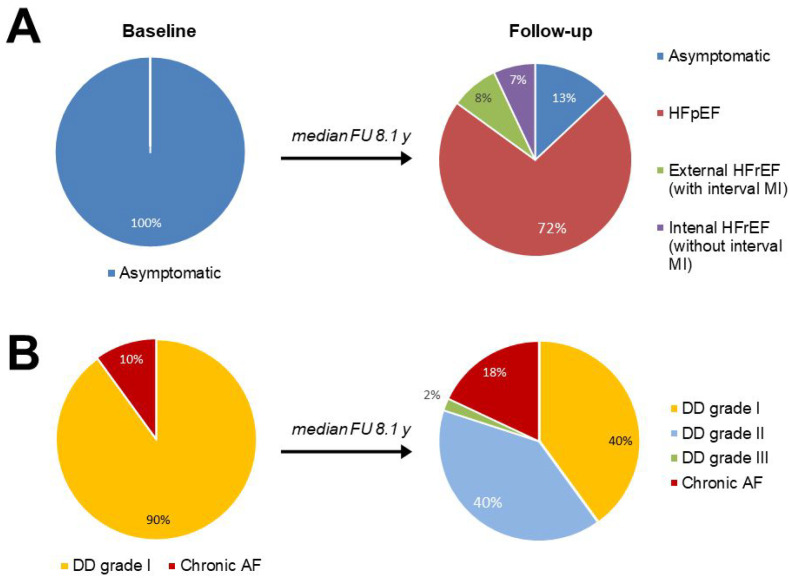
Changes in clinical status (**Panel A**) and LV diastolic dysfunction (DD) grade and atrial fibrillation (AF) incidence (**Panel B**) in patients with concentric asymptomatic LVH after a median follow-up (FU) of 8.1 years.

**Table 1 jcm-11-03885-t001:** Clinical and echocardiographic characteristics in patients with hypertensive concentric LV hypertrophy during the follow-up.

Variables	Hypertensive LV Hypertrophy		*p* Value
	*n* = 223		
	Baseline	Follow-Up	
Clinical parameters:			
Age, y	59 (52–68)		
Men	65%		
Duration of hypertension, y	20 (11–30)		
Hypertension ^a^	100%	100%	1
Chronic atrial fibrillation	10%	18%	0.027
Ischemic heart disease	35%	44%	0.053
Myocardial infarction	18%	28%	0.01
Myocardial revascularization	14%	19%	0.097
Diabetes mellitus	20%	31%	0.009
Body mass index, kg/m^2^	30.4 (27.2–34.4)	30.8 (27.8–35.0)	0.06
Obesity ^b^	52%	57%	0.34
Chronic kidney disease ^c^	29%	40%	0.013
Chronic obstructive pulmonary disease	9%	13%	0.22
Clinical status of LV hypertrophy:			
Asymptomatic	100%	13%	<0.001
HFpEF	0%	72%	
HFrEF	0%	15%	
A follow-up (interval) therapy:			
ACEI/ARB		66%	
β-Blockers		56%	
Calcium channel blockers		24%	
Loop diuretics		43%	
Thiazide diuretics		23%	
Mineralocorticoid receptors antagonists		9%	
Statins		40%	
≥2 antihypertensive drugs		68%	
Echocardiographic measures			
LV mass index, g/m^2^	136 (116–158)	136 (116–171)	0.17
LV end-diastolic dimension, cm	5.2 (5.0–5.6)	4.8 (4.5–5.0)	<0.001
Relative wall thickness	0.46 (0.44–0.49)	0.54 (0.51–0.60)	<0.001
Type of LV hypertrophy:			0.061
Concentric	100%	98%	
Eccentric	0%	2%	
LA anteroposterior diameter, cm	4.1 (3.9–4.4)	4.5 (4.2–4.9)	<0.001
LV diastolic dysfunction,^d^ grade:			<0.001
I	100%	48%	
II	0%	49%	
III	0%	3%	
Pulmonary artery systolic pressure, mm Hg	24 (23–30)	32 (27–40)	<0.001
Pulmonary hypertension,^e^ *n* (%)	13%	40%	<0.001

Data are presented as the median (interquartile range) for continuous variables, and percent for categorical variables. ^a^—blood pressure ≥ 140/90 Hg mm; ^b^—body mass index ≥ 30 kg/m^2^; ^c^—estimated glomerular filtration rate < 60 mL/min/1.73 m^2^; ^d^—among patients with sinus rhythm; ^e^—pulmonary artery systolic pressure > 35 mm Hg. ACEI indicates angiotensin-converting enzyme inhibitor; ARB, angiotensin receptor blocker; HFpEF, heart failure with preserved ejection fraction; HFrEF, heart failure with reduced ejection fraction; LV, left ventricular; LA, left atrial.

**Table 2 jcm-11-03885-t002:** Clinical variables and therapy in new-onset HF patients vs. asymptomatic patients.

Variables	Asymptomatic (*n* = 28)		Transition to HFpEF (*n* = 161)		Transition to Internal HFrEF (*n* = 16)	
	Baseline	Follow-Up	Initial Visit	Follow-Up Visit	Initial Visit	Follow-Up Visit
Clinical parameters:						
Age, y	54 (48–64)		61 (52–69) ^§^		58 (54–63)	
Men	57%		62%		81%	
Duration of follow-up, y		8.2 (6.4–9.6)		8.1 (6.9–10.2)		9.1 (6.9–11.4)
Duration of hypertension, y	15 (10–20)		20 (14.5–30) ^§^		25 (10–30) ^§^	
Chronic atrial fibrillation	0%	0%	8%	15% ^*§^	38% ^§§µµ^	63% ^§§µµ^
Ischemic heart disease	29%	32%	37%	43%	6% ^µ^	13% ^µ^
Previous MI	14%	25%	21%	23%	6%	6%
Myocardial revascularization	11%	18%	15%	19%	6%	6%
Diabetes mellitus	14%	25%	21%	31%	19%	44%
Body mass index, kg/m^2^	29.8 (26.4–34.6)	30.8 (27.8–35.9)	30.0 (27.5–34.0)	30.5 (28.0–33.8)	32.2 (29.7–38.0)	35.4 (29.3–40.1) ^µ^*
Obesity ^a^	50%	57%	50%	54%	75%	75%
Chronic kidney disease ^b^	18%	29%	33%	44%	25%	44%
COPD	14%	18%	6%	9%	19%	25%
Systolic BP, mm Hg	138 (123–159)	138 (122–156)	148 (130–159)	149 (131–161)	152 (138–159)	154 (139–160)
Diastolic BP, mm Hg	85 (81–102)	83 (78–101)	85 (82–102)	85 (83–102)	94 (85–102)	94 (86–103)
Heart rate, bpm	67 (60–76)	65 (60–76)	69 (64–77)	65 (60–73) **	72 (70–78)	78 (70–90) *^§§µµ^
LDL-cholesterol, mmol/L	–	3.1 (2.8–3.5)	–	2.9 (2.4–3.3)	–	3.1 (2.3–3.3)
Interval therapy (between the baseline and follow-up studies)						
ACEI/ARB	–	79%	–	61%	–	81%
β-Blockers	–	64%	–	53%	–	63%
Calcium channel blockers	–	21%	–	24%	–	25%
Loop diuretics	–	0%	–	47% ^§§^	–	69% ^§§^
Thiazide diuretics	–	32%	–	21%	–	38%
MRA	–	7%	–	9%	–	13%
Statins	–	57%	–	34% ^§^	–	13% ^§§^
Number of antihypertensive drugs, *n*	–	2.0 (1.5–3.0)	–	2.0 (1.0–3.0)	–	2.5 (2.0–3.5)

Data are presented as the median (interquartile range) for continuous variables, and percent for categorical variables. ^a^—body mass index ≥ 30 kg/m^2^; ^b^—estimated glomerular filtration rate < 60 mL/min/1.73 m^2^; ACEI indicates angiotensin-converting enzyme inhibitor; ARB, angiotensin receptor blocker; BP, blood pressure; CI, confidence interval; COPD, chronic pulmonary obstructive disease; HFpEF, heart failure with preserved ejection fraction; HFrEF, heart failure with reduced ejection fraction; LDL, low-density lipoprotein; LV, left ventricular; MRA, mineralocorticoid receptors antagonists; NT-proBNP, N-terminal pro–brain natriuretic peptide. * *p* < 0.05, ** *p* < 0.01 vs. baseline. ^§^ *p* < 0.05, ^§§^ *p* < 0.01 vs. asymptomatic patients. ^µ^
*p* < 0.05, ^µµ^
*p* < 0.01 vs. patients with HFpEF.

**Table 3 jcm-11-03885-t003:** Echocardiographic variables in new-onset HF patients vs. asymptomatic patients.

Variables	Asymptomatic (*n* = 28)		Transition to HFpEF (*n* = 161)		Transition to Internal HFrEF (*n* = 16)	
	Baseline	Δ from Baseline (95% CI)	Baseline	Δ from Baseline (95% CI)	Baseline	Δ from Baseline (95% CI)
LV mass index, g/m^2^	118 (112–142)	−13 (−26, −1) *	136 (116–160)	6 (−0.1, 12) ^§§^	151 (122–158)	6 (−6, 21) ^§^
LV end-diastolic dimension, cm	5.1 (4.9–5.3)	−0.5 (−0.7, −0.4) **	5.3 (4.9–5.5)	−0.6 (−0.6, −0.5) **	5.7 (5.4–5.8) ^§§µµ^	−0.4 (−0.7, −0.1) **
Relative wall thickness	0.48 (0.43–0.51)	0.07 (0.05, 0.09) **	0.46 (0.44–0.49)	0.10 (0.09–0.11) **^§^	0.44 (0.43–0.45) ^§µµ^	0.07 (0.02–0.11) **
LV ejection fraction, %	61 (56–65)	2 (−1, 5)	60 (58–64)	2 (−1, 4)	58 (54–60) ^§µµ^	−13 (−16, −10) **^§§µµ^
LA anteroposterior diameter, cm	4.0 (3.8–4.0)	0.3 (0.2, 0.35) **	4.1 (3.9–4.3) ^§§^	0.45 (0.35, 0.5) **^§^	4.5 (4.4–4.7) ^§§µµ^	0.6 (0.3–0.9) **^§^
Progression of LVDD ^a^		0		85 (62) ^§§^		4 (67) ^§§^
PASP, mm Hg	25 (23–30)	0 (−3, 4)	25 (23–29)	8 (6, 10) **^§§^	24 (23–38)	13 (7, 20) **^§§^
Pulmonary hypertension ^b^	4%	7%	12%	42% **^§§^	25%^§^	69% **^§§µ^
Mitral E/e′ ratio ^c^		8.2 (7.5–9.1)		11.9 (10.3–14.3) ^§§^		11.9 (11.2–13.4) ^§§µµ^
LA volume index, mL/m^2^, ^c^		32 (28–37)		41 (37–50) ^§§^		54 (40–63) ^§§µµ^

Data are presented as the median (interquartile range) for continuous variables, and percent for categorical variables. ^a^—an increase in DD grade from I to II-II; ^b^—pulmonary artery systolic pressure > 35 mm Hg; ^c^—data are presented as the median (interquartile range). CI indicates confidence interval; E, early inflow velocity; e′, averaged annulus relaxation velocity; DD, diastolic dysfunction; HFpEF, heart failure with preserved ejection fraction; HFrEF, heart failure with reduced ejection fraction; LV, left ventricular; PASP, pulmonary artery systolic pressure. * *p* < 0.05, ** *p* < 0.01 vs. baseline. ^§^ *p* < 0.05, ^§§^ *p* < 0,01 vs. asymptomatic patients. ^µ^
*p* < 0.05, ^µµ^
*p* < 0.01 vs. patients with HFpEF.

**Table 4 jcm-11-03885-t004:** Independent predictors of HFpEF in asymptomatic patients with concentric LV hypertrophy.

Variables	Coefficient	Standard Error	Odds Ratio	95% CI	*p* Value
Interval statin treatment	−1.165	0.456	0.31	0.128 to 0.762	0.011
Duration of hypertension	0.048	0.023	1.05	1.001 to 1.099	0.044
Age	0.045	0.022	1.047	1.003 to 1.094	0.039
Interval change in LV mass index	0.017	0.006	1.017	1.005 to 1.029	0.006

CI indicates confidence interval; LV, left ventricular.

**Table 5 jcm-11-03885-t005:** Independent predictors of internal HFrEF in asymptomatic patients with hypertensive concentric LVH.

Variables	Coefficient	Standard Error	Odds Ratio	95% CI	*p* Value
Interval chronic atrial fibrillation	1.856	0.885	6.40	1.33 to 17.6	0.017
Baseline LV end-diastolic dimension	1.575	0.660	4.84	1.13 to 36.3	0.036
Baseline LV ejection fraction	−0.175	0.0818	0.84	0.72 to 0.99	0.032

CI indicates confidence interval; LV, left ventricular.

## Data Availability

The authors confirm that the data supporting the findings of this study are available within the article. Raw data that support the findings of this study are available from the corresponding author, upon reasonable request.

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
