# Peer review of "Asymptomatic Left Ventricular Hypertrophy Is a Potent Risk Factor for the Development of HFpEF but Not HFrEF: Results of a Retrospective Cohort Study"

_jcm, 2022, doi:10.3390/jcm11133885_

Round 1

Reviewer 1 Report

I read with interest the article entitled „Intravascular lithotripsy during the percutaneous coronary intervention: current concepts ” which concerns an important and clinically interesting issue. The manuscript contains many valuable comments and information and is generally well-written. However, I have several minor comments regarding this manuscript.

1-    Introduction section LVH is not only connected with HA, (for example athletes, CKD, toxic; congenital etc) I suggest adding a short comment on this topic.

2-    Material and Methods section please provide exactly number of approved by the Ethic Committee and it full name.

3-    Material and Methods section- Despite study is really interesting it might be underpowered- had authors made a power analysis ?

4-    Material and Methods section-27 patients died before finishing follow up – this group is really interesting – are there any data regarding reasons of death in this subpopulation, please make a short comments on this issue in manuscript (discussion section)

5-    Discussion is really well written

6-    In my opinion short paragraph  gathering all of study limitations at the end of discussion section is missing.

Reviewer 2 Report

Authors reported the outcomes of asymptomatic patients with left ventricular hypertrophy (LVH) for more than 8 years and elucidated the predictors of occurrence of heart failure with preserved ejection fraction (HFpEF) and with reduced ejection fraction (HFrEF). It is a very important topic and the presented predictors were convincing.

Major findings

Were patients with left ventricular concentric remodeling included in this study?

 Among 350 patients, 27 patients died and they were excluded from this study. What were the causes of their death? Was there anyone who had dead due to heart failure?  Authors wrote that all of these patients were with more than mild LVH. So, some patients seemed to be dead due to cardia event. These patients may be analyzed as patients with cardiac events.

 Authors described the importance of statin to prevent the occurrence of HFpEF. How about the proportion of patients with dyslipidemia in each group? What were the reasons or indications of prescribing stain for the patients in each group during the follow-up period? 

Minor finding

In Table 2

Incidence of ischemic heart disease in patients with transition to internal HFrEF may be misspelled.
